# Microglia-Mediated Neurodegeneration in Perinatal Brain Injuries

**DOI:** 10.3390/biom11010099

**Published:** 2021-01-13

**Authors:** Bobbi Fleiss, Juliette Van Steenwinckel, Cindy Bokobza, Isabelle K. Shearer, Emily Ross-Munro, Pierre Gressens

**Affiliations:** 1School of Health and Biomedical Sciences, RMIT University, Bundoora, VIC 3085, Australia; bobbi.fleiss@rmit.edu.au (B.F.); s3741737@student.rmit.edu.au (I.K.S.); emily.ross-munro@rmit.edu.au (E.R.-M.); 2NeuroDiderot, Inserm, Université de Paris, Paris 75019, France; juliette.van-steenwinckel@inserm.fr (J.V.S.); bokobzacindy@gmail.com (C.B.)

**Keywords:** neuroprotection, encephalopathy, neuroinflammation, stroke, prematurity, neurodegenerative disorders

## Abstract

Perinatal brain injuries, including encephalopathy related to fetal growth restriction, encephalopathy of prematurity, neonatal encephalopathy of the term neonate, and neonatal stroke, are a major cause of neurodevelopmental disorders. They trigger cellular and molecular cascades that lead in many cases to permanent motor, cognitive, and/or behavioral deficits. Damage includes neuronal degeneration, selective loss of subclasses of interneurons, blocked maturation of oligodendrocyte progenitor cells leading to dysmyelination, axonopathy and very likely synaptopathy, leading to impaired connectivity. The nature and severity of changes vary according to the type and severity of insult and maturation stage of the brain. Microglial activation has been demonstrated almost ubiquitously in perinatal brain injuries and these responses are key cell orchestrators of brain pathology but also attempts at repair. These divergent roles are facilitated by a diverse suite of transcriptional profiles and through a complex dialogue with other brain cell types. Adding to the complexity of understanding microglia and how to modulate them to protect the brain is that these cells have their own developmental stages, enabling them to be key participants in brain building. Of note, not only do microglia help build the brain and respond to brain injury, but they are a key cell in the transduction of systemic inflammation into neuroinflammation. Systemic inflammatory exposure is a key risk factor for poor neurodevelopmental outcomes in preterm born infants. Based on these observations, microglia appear as a key cell target for neuroprotection in perinatal brain injuries. Numerous strategies have been developed experimentally to modulate microglia and attenuate brain injury based on these strong supporting data and we will summarize these.

## 1. Perinatal Brain Disorders

Perinatal brain injuries (PBIs) represent a number of entities of different origins that are each responsible for causing neurodevelopmental disorders (NDDs). The present review will focus on four causes of PBI that are highly prevalent both in westernized and emerging countries: (i) encephalopathy associated with fetal growth restriction (EoFGR); (ii) encephalopathy of prematurity (EoP); (iii) neonatal encephalopathy of term neonates (NE), and; (iv) neonatal stroke (NS) that can occur both in preterm and term neonates. 

Although PBIs are not typically considered to be neurodegenerative disorders, we would like to highlight that they all involve to a greater or lesser extent changes to neurons, that in the case of EoFGR, NE and NS certainly involve cell death [1,2,3] and for EoP that involves dysfunction related to neuronal maturation and sub-cellular organization/connectivity [4,5,6]. Specifically, it has been shown that there is a link between being born preterm and cognitive decline associated with Alzheimer’s disease [7], highlighting that neurodegenerative disorders are truly life-span conditions. Persisting effects of perinatal brain injuries such as this are increasingly recognized [8,9,10,11,12], leading to increased risks of other neurological injuries and disorders [13,14,15,16], (and reviewed in [17]). In addition, although it seems that the ‘holy grail’ in adult neurodegeneration is the controlled recapitulation of developmental steps to regenerate the brain, it is striking that the still developing brain is not capable of this redeployment of developmental processes. As such, lessons learnt from the similarities and differences between neurodegenerative events early versus late in life will be valuable to us all in the hunt for the master switch of repair. 

For recent reviews on the epidemiology, etiology, radiological and clinical spectrum of the described PBIs, please refer to [18,19,20,21,22]. In brief, these are frequent brain disorders: 3–9% neonates are born with FGR (up to 30% in low-resource settings), 6–11% neonates are born preterm (below 37 weeks of gestation), 1–3.5/1000 term neonates develop NE, and NS occur in at least 1/4000 neonates. Major negative outcomes include motor deficits (including cerebral palsy), cognitive impairment, and increased risk for several behavioral deficits including autism spectrum disorders (ASD), attention deficit hyperactivity disorder (ADHD), schizophrenia, psychosis, depression, and bipolar disorder [19,20,23,24,25]. The most affected brain structures are briefly summarized in Table 1. Of note, especially in the more diffuse forms of encephalopathy (EoP, EoFGR), other brain structures not mentioned in Table 1 might also be significantly affected but have not been studied in a such detailed and/or repeated manner to date.

## 2. Origins, Development and Heterogeneity of Microglia

Microglia (MG) are currently the only known immune cells which permanently reside in the CNS and they make up approximately 10% to 15% of the cells in the brain, depending on the region [26]. The roles of MG can be categorised into two general functions: (1) innate immune defense, including responding to invading pathogens or clearing away debris from injury, and (2) development and homeostatic roles which include supporting the formation of cortical lamination, oligodendrocyte maturation and facilitating learning and memory acquisition via phagocytosis. For in-depth reviews of the roles of MG in development please see [27,28].

MG are derived from the myeloid cell linage and arise from yolk sack (YS) primitive macrophages and possibly also progenitor cells derived from hematopoietic cells [29], but this is still an issue of some debate. MG invasion occurs in two waves (data predominantly from the mouse) with the first wave occurring between embryonic day (E) 8.5 and E14.5 in the mouse. Specifically, the E8.5 mouse embryo develops erythromyeloid progenitors within the yolk sack [30]. A subset of these erythromyeloid progenitors, regulated by the expression of PU1, RUNX1 and IRF8, give rise to MG progenitors [31]. At E9.5 to E10.5, these MG progenitors migrate to the central nervous system (CNS) where they distribute and continue to proliferate [32,33]. The peripheral hematopoietic system continues to supply MG progenitors until the formation of the blood brain barrier at E15.5 [34]. After this event the MG population in the brain is fixed and becomes self-sustaining [17,18]. Once MG are established in the brain, approximately 4.5 week’s human gestation, they express ionized calcium-binding adaptor protein 1 (Iba1) and a suit of additional markers indicated in Figure 1.

The second wave of MG invasion is marked by a rapid increase in numbers between E14 and E17 [41]. Since the abrupt and rapid increase in MG density cannot be explained by proliferation alone, it was concluded that there is a second MG invasion following development of the brain vasculature [42]. However, it is yet to be determined if the second wave of MG also originated from the yolk sac, whether they follow a different lineage to the first wave, or if there is a correlation between endothelial cells and second wave MG. Furthermore, it is unknown if MG from the first wave are phenotypically, developmentally and functionally different to MG from the first wave. Another important point to note, is that the invasion and proliferation of MG in human [43], mouse [44], and sheep (Shearer and Fleiss, unpublished observations) occurs via the subventricular zones, with MG accumulating and sitting adjacent to oligodendrocyte progenitor cells (OPCs). This unfortunate juxtaposition makes the OPCs particularly vulnerable to soluble products released from the MG activated by injury or infection.

Once in the brain, MG pass through three temporally regulated stages of development: early MG, pre-MG, and adult MG (Figure 1). Each of these three stages are marked by changes in gene expression, morphology, and the specific roles that these cells undertake [36,45,46,47]. Early MG display increases in cell cycling and chromatin remodeling genes, including Dab2, Mcm5 and Lyz2 [35]. Two early MG markers, Ifit3 and Dnmt1, which have also been identified in mouse have also been detected in pre-MG from the neonatal pig brain [48]. Towards the end of mouse gestation (approximately equivalent to the end of the second trimester in the human) there is a distinct change in MG gene profile. This is marked by the down regulation of cell differentiation genes observed in early MG, and the up regulation of genes associated with synaptic pruning which is a hallmark of the pre-MG stage. The maturation stage dependent processes during the pre-MG stage also include controlling the death of unnecessary progenitors [49], regulating cortical layering [44], and as mentioned synaptic pruning [50]. The specific mechanism controlling MG development are unknown, but studies focused on epigenetic suggest these processes play a key role, reviewed in [51]. Of note, using ATAC-seq that identifies accessible regions within promoters (H3K4me3+ regions near the transcription start site of genes), substantial shifts in the chromatin conformation between early MG and adult MG were identified [35]. However, there were no apparent changes in early MG suggesting that transcriptional changes in early MG my not be mediated by chromatic changes, but by alternative regulators of transcriptional activity. As the mouse reaches adulthood, the pre-MG genetic profile is downregulated. Simultaneously, genes associated with immune surveillance (including Cd14, MafB and Met2a) are up regulated [35]. This final stage in maturation is estimated to occur at 14 years in the human (Figure 1).

MG populations in different regions of the brain vary in terms of gene expression [52,53,54], density [55] and morphology [56]. Of note, MG abundance in the cortex and hippocampus is two-fold higher than in the thalamus and midbrain [55] and MG populations also vary between layers of highly structured regions, including within the cortices and the hippocampus [57]. Furthermore, mouse MG are present at higher density in the grey matter compared to white matter [55]. MG branching also differ between the grey and white matter: white matter MG typically being bipolar, whereas there is far more complex branching in the cortices, striatum and [58]. Another critical mediator of differences in MG state is sex [59,60,61], and this is now hypothesized to underly critical differences in disease progression in adult neurodegeneration [62] and also likely explains sex differences in PBI and models of injury [63]. Of note, MG isolated from adult mice retain their sex specific transcriptomic phenotype even when transplanted in hosts of the opposite sex [64] and male sex is associated with a slower maturation of MG [65], with females having significantly increased maturity of pathways such as for modulating cytokine production and mast cell and leukocyte activation. Species is also another significant factor in the expression profile of MG [66] and a focus on understanding changes the core across-species markers will improve the translational potential of work on MG. 

## 3. Role of Systemic Inflammation in PBIs

Human clinical and epidemiological studies strongly argue in favor of a major role of a sustained systemic inflammation in the pathophysiology of several PBIs, reviewed in [21,67,68]. This has been particularly addressed in the field of EoP [69]. Indeed, preterm infants (and especially very and extremely preterm infants) are at high risk to be exposed, before, during and after birth, to several conditions that lead to systemic inflammation as measured by increased levels of circulating pro-inflammatory cytokines [70], including chorioamnionitis and funisitis, peri-partum hypoxic/hypoxic-ischemic insults including intermittent hypoxemia, postnatal lung mechanical ventilation, sepsis or necrotizing enterocolitis (Figure 2).

Animal models have shown that systemic inflammation in itself can impact the developing brain. Alternatively, in the so-called multiple hit hypothesis (Figure 3), systemic inflammation can act as a predisposing factor making the brain more susceptible to a second stress (sensitization process). Indeed, injection of low doses lipopolysaccharide (LPS, a TLR4 agonist derived from *E. coli*) to developing rats makes the newborn brain significantly more susceptible to hypoxic-ischemic insult [74]. Similarly, injection of the prototypical pro-inflammatory cytokine interleukin-1-beta (IL-1β) to newborn mice or rats increases sensitivity to an excitotoxic insult [75,76]. The mechanisms mediating sensitization are not yet fully understood, however, could include changes in gene transcription in MG and modifications of metabotropic glutamate receptor activity in neurons, potentially through changes in GRK2 [17,18,19,20,21,77]. From a clinical point of view, a sensitizing effect of systemic inflammation linked to *in utero* infection, has been suggested in some term infants with NE that is resistant to hypothermia [78]. It is important to note, that evidence accrues that the nature of the infectious agent is critical for determining the specific pattern of systemic response and whether brain injury occurs [79,80].

For a long time, the brain was thought to be isolated from the rest of the body, protected behind the blood-brain barrier (BBB) and CSF-brain barrier. On the other hand, textbooks informed us that the immature brain, including the neonate one, had immature barriers that left the brain vulnerable. As often, these over-simplified dogmas proved to be incorrect [82]: (i) there is an important cross talk between the CNS and the periphery, in particular the gut and its microbiota, through circulating factors and the vagal nerve, among others [83]; and (ii) the BBB matures quite early during brain development and is fully mature in neonates and even more resistant to some insults (such as ischemic stroke) when compared to adult BBB [84]. Of note, MG protect neurovascular integrity following NS in the P7 rat, as prior clodronate-liposomal depletion of MG induced hemorrhages in the injured area when assessed at 24 h post-injury [85]. Whether this protective effect is found in other models of PBI has yet to be determined.

An area of interest for integrating the ideas of systemic inflammation, effects of the brain development and common disorders related to PBIs such as ASD is the gut-brain axis [86]. Although the impact on brain maturation and future function is not yet known, preterm and FGR infants have been shown to have an abnormal microbiota when compared to term healthy controls [87]. In addition, recent studies suggest that this abnormal microbiota could be long-lasting [88]. There are also significant changes in the gut structure and function associated with gene defects identified in people with ASD [89] Further integrative studies, both in humans and relevant animal models, of the interactions between the immune system, the gut, the microbiota, and the developing brain appear of paramount importance. Additional studies will allow us to obtain in an in-depth understanding of the potential disturbed cross-talks between the brain and the periphery in the genesis of PBI’s and their morbid consequences on brain functions. 

## 4. Role of Microglia in PBIs

Human post-mortem and/or experimental studies have highlighted MG activation as a key early hallmark of PBI [72,90,91,92,93,94]. Studies of animal models of EoP (and, to a lesser extent, of EoFGR) have mostly focused on innate immune cells, showing a strong and predominant involvement of MG with some transient recruitment of small numbers of peripheral neutrophils, monocytes, and macrophages. The contributing role of these latter innate immune cells to brain pathology has been so far poorly explored. This is likely due to papers reporting only limited roles for cells, such as macrophage in perinatal brain injury [94,95] although neutrophils and T cells have been implicated after HI [96,97]. Studies have also been hampered by a lack of specific tools to discriminate MG and macrophage (until relatively recently, see [98,99]). Similarly, the potential contribution of perivascular macrophages is far from being elucidated [100], although there is tantalizing evidence that perinatal environmental factors such as maternal high-fat diet during gestation and lactation changes these cells and sensitizes them to β-amyloid accumulation [101]. In pre-clinical models of NE, time profiles of innate and adaptive cell recruitment at the site of the lesion have been characterized (Figure 4), highlighting the potential roles of multiple innate cell types in progression of brain damage. Of note, when compared to models of EoP, the NE models in which these studies are based are much more severe insults/brain damage and are performed in a more mature animal (modeling the preterm vs term infant). The overwhelming body of evidence to data states that the ‘activation’ of MG after injury is damaging, supported by studies in which ablating MG or applying blanket suppression of their functions improves outcomes (reviewed in [90,102]). However, studies do report protective roles for MG [85,103] and the potential for endogenous positive roles (that may be valuable neurotherapeutics themselves) is increasingly supported by studies that look over time at the role for these cells [83,97,98]. Blanket suppression or ablation of MG has also failed to show persisting efficacy [104] or to worsen outcomes [105,106] in models of injury to the immature brain and in adult neurodegenerative disorders [107] and models of injury [108,109] further highlighting that MG are likely not the ‘bad guys’ all the time.

A limited number of experimental studies have aimed at deciphering the molecular pathways underlying MG activation for the goal of down-regulating damaging pathways and also potentially supporting MG to acquire a beneficial phenotype in terms of brain cleaning, repair, and plasticity [110]. Most of these studies have employed gene expression profiling in cell-sorted MG—using various approaches and markers to determine the specificity of their cell populations. We will mention that studies have shown in experimental model of EoP and in preterm infants with abnormal white matter, the identification of lipid metabolism pathways including the PPARγ pathway [111,112], Wnt pathway inhibition [91], and DLG4/PSD-95 [94]. In addition, there is a link between MG activation and EoP phenotype for microRNA (miR)-146b in an experimental model of EoP, and the oxytocin receptor-mediated pathway in an experimental model of EoFGR [113]. The identification specifically in MG of PSD-95 in a stage of brain development during which preterm birth is reasonably common (22–33 weeks) and when PSD-95 is not yet expressed in neurons is of particular interest. PSD-95 mutations have been found in some cases of ASD, childhood epilepsy, and intellectual deficiency, raising the question of the role of abnormal MG in these disorders that have been viewed as “neuronal/synaptic disorders”. In addition, these studies have shown that MG not only become potently pro-inflammatory even in the context of a mild/moderate systemic inflammatory context, but down-regulate most of the pathways related to normal functions of MG during brain development [94]. This supports the concept that some PBIs induce a brain maldevelopment by disrupting normal brain development programming [114,115].

Our recent studies have also highlighted the fact that MG genome expression remained disturbed weeks and months (and we suspect even years) beyond the perinatal systemic inflammation insult [94]. This is also supported by the excellent work of others across models of PBI (and in models of maternal immune activation) that there are long-term changes into MG biology following an insult [116,117,118,119,120] and that this might be phase of injury that can be targeted to improve outcomes [117,121]. This observation, combined with clinical and experimental data in adult brain disorders such as TBI [122], has led to the hypothesis of a tertiary phase of PBIs [17,123] that were until recently viewed as non-progressive disorders [94,124] (Figure 5).

Tertiary changes are in addition to developmental disruption associated with the initial insult to the immature brain and reflect injury processes can persist for many months or years, such as persistently activated MG, potentially related to epigenetic changes (reviewed in the context of perinatal stress in [125]). These processes are implicit in prevention of endogenous repair and regeneration and predispose patients to development of future cognitive dysfunction and sensitization to further injury.

## 5. Limitations of Microglial Studies in the Immature Animal

Different approaches have been used to ablate MG and hence demonstrate their role in neuropathology in animal models. However, when applied to newborn animals, all these approaches have limitations that need to be considered when interpreting the data: (i) embryonic depletion of MG (i.e., PU1 or CX3CR1 KO) itself alters brain development [126] in ways that could alter the experimental paradigm; (ii) postnatal ablation of MG with a Tamoxifen driven transgenic or other pharmacologic approaches (i.e. PLX3397, ganciclovir) require approximately 3 days to be efficient, making them inappropriate when targeting the first postnatal days in the experimental model and prenatal tamoxifen increases pup mortality and requires cesarean delivery and cross fostering [127] processes also known to alter experimental outcomes; in addition, in our hands PLX3397 (3 or 30 mg/kg in vivo or 1 µM in vitro) induced a reduced expression of several MG genes (including *Iba1* and *Cx3cr1*) without affecting MG density [50] indicating that care must be taken to ensure ablation and not an incomplete silencing, (iii) gadolinium chloride (GdCl3), which kills pro-inflammatory MG via competitive inhibition of calcium mobilization and damage to the plasma membrane, has to be injected intracerebrally (as it does not cross the intact BBB) and only has a focal effect on MG [91,128] (iv) intracerebral injection of liposome-encapsulated clodronate [103], which has similar advantages and limitations as GdCl3.

The remaining part of this review will focus on the potential role of MG on different brain cell types involved in PBIs. Although we believe MG are a key player in these disorders, one should not ignore the potential roles of other cells types (beyond other immune cells already mentioned above) in the pathophysiology of PBIs.

## 6. Microglia and Other Cellular Brain Partners

### 6.1. Microglia Effects on Oligodendrocyte Progenitor Cell Maturation and Myelin Deficit in PBIs

Human postmortem studies of EoP and animal models of EoP and EoFGR have identified an oligodendrocyte progenitor cell (OPC) maturation arrest, leading to dysmyelination [92,129]. One experimental study, using GdCl3 to ablate pro-inflammatory MG, has demonstrated a direct role of MG in the white matter phenotype in a mouse model of EoP [91], and previous work has used antibody and pharmacologic approaches to show a similar causal role for MG in perinatal excitotoxic lesion development [130]. In addition, a specific role for a subset of MG, CD11c-ve, has been demonstrated by studying responses to injury and development using ablation and gene profiling [131]. This study revealed that this MG subset produced factors, including IGF-1, that are necessary for myelination and neurogenesis. Additional studies will be important to generalize this concept of MG-induced OPC maturation arrest in PBIs.

### 6.2. Microglia Interaction with Astrocytes in PBIs

Several post-mortem and preclinical studies have highlighted the importance of astrocytes and their potential interactions with MG [60,61]. These include a study that demonstrated using transgenic, and pharmacologic approaches in vivo and in vitro that astrocyte derived IL33 is an endogenous neuroprotective agent in a mouse more of NE including by HI including by supporting the M2-like phenotype of MG [132]. In an EoP model, the existence of a tripartite glial circuits has been demonstrated: the OPC maturation blockade by MG activation required an astrocytic reactivity, via an overproduction of cyclo-oxygenase-2 (COX)-2 [133]. It is interesting to highlight that reactive astrocytes have been associated to adult neurodegenerative diseases as Alzheimer’s, Parkinson’s and Multiple Sclerosis [134]. In the future, it will be pertinent to study reactive astrocyte profile in PBIs models and correlate it (or not) to the one observed in adult models.

### 6.3. Microglia Effects on Neuronal Cell Death in PBIs

In term NE and NS, neuronal cell death is a key feature. The absence of frank levels of cell death in contemporaneous cohorts of EoP is reviewed in detail in [135]. Human post-mortem studies of term NE cases and animal models of NE (hypoxic-ischemic insult) and NS (middle cerebral artery occlusion) have identified different types of neuronal cell death: necrosis, apoptosis, excitotoxic cell death, necroptosis and autophagy [136,137] and review in [138,139]. When studied over time, neurodegeneration has been shown to last for several days (and up to a couple of weeks) after the experimental insult [140]. This delayed cell death is generally referred to the ‘secondary phase’ but the ‘tertiary phase’ is probably a more accurate description of death occurring past one week—as it likely is not linked directly to the insult. Although this neurodegeneration is accompanied by a protracted inflammatory response as described above (Figure 4), to our knowledge, only two studies addressed directly the role of MG in neuronal cell death. In a model of transient middle cerebral artery occlusion in P7 rats, ablation of MG with intracerebral injection of liposome-encapsulated clodronate increased the brain levels of several cytokines and chemokines and increased the severity and volume of injury, suggesting that MG contribute to endogenous protection during the subacute injury phase [103]. In contrast, partial ablation of MG with anti-MAC1 antibody coupled to the toxin saporin significantly reduced neuronal cell death in a model of neonatal excitotoxic brain lesion [141]. These studies suggest that the impact of MG on neuronal cell death might be dependent upon the injury stimulus and/or the timing after the insult, in agreement with the fact that MG can adopt different phenotypes that can be either deleterious or protective for the injured developing brain [142]. In the context of normal development, MG are known to directly and specifically mediate the apoptotic cell death of retinal cells via there production of nerve growth factor (NGF) [143] and Layer V cortical neurons require trophic support for survival from MG, in the form of MG-secreted IGF1 [44].

### 6.4. Microglia Effects on Interneurons in PBIs

Interneurons originally appear in the subpallium before tangentially migrating through the neocortex, to then emerge radially through the cortical plate (CP) and distribute throughout cortical laminae [144]. In the mouse, interneuron development continues during the first 3 weeks of postnatal life, and in the human this process continues through the third trimester and early neonatal life making interneurons particularly vulnerable to disturbance from preterm birth. Recent human postmortem studies performed in preterm infants have shown a deficit in some subpopulations of interneurons, in particular, cortical parvalbumin (PV) positive interneurons [6,145]. Such a selective deficit of interneurons has also been described in mouse models, a rat model and a sheep model of EoP [6,146,147]. The precise origin of the deficits are still unknown and could be due to impaired production, migration, survival, localisation and/or differentiation of these interneurons.

Importantly, interneuron migration into the CP occurs on a timeline parallel with MG invasion of the CP. In a mouse depleted of MG (*Cx3cr1*^–/–^) and a model of maternal immune activation, interneurons prematurely enter the CP [126], and when assessed at P7, an increased number of interneurons were found in the upper cortical layers. By P20, there were increased PV+ interneurons within layers III/IV, and so MG have been proposed to mediate development of cortical interneurons [148]. Early aberrant MG activation in cases of PBI may interfere with these cellular processes [141]. A recent interesting study investigated the specific effects on MG on interneurons derived from iPSC [149]. This novel study identified that inflammatory-activated MG induced profound changes in the interneurons including compromising energy-dependent neuronal processes such as arborization, synapse formation and synaptic GABA release.

### 6.5. Microglia Effects on Neuronal Connectivity in PBIs

MRI studies have identified connectivity deficits in preterm infants compared to term infants [150,151]. Using ultrafast Doppler in a model of EoFGR it has been beautifully shown that prenatal malnutrition linked brain injury causes significant changes in connectivity and this is rescued by neuroprotective agent targeting MG activation [113]. However, to the best of our knowledge, connectivity has not yet been fully investigated across animal models of PBIs, not allowing yet to determine experimentally the potential role of MG in this connectivity deficit. However, MG engulf synapses in the healthy brain to refine circuitry during learning in the adult [152] and during development the absence of MG (Cx3cr1 KO) leads to disturbances in connectivity [50].

## 7. Microglia, a Target for Neuroprotection in PBIs?

Several studies have been focusing on targeting ‘activated’ MG in order to protect the developing brain against perinatal insults or stimulate repair (Table 2). One potential limitation of these studies is that, in most instances, the molecular target is not specific to MG. However, advances in bioengineering has led to a few studies, using either liposome-based delivery or nanoparticles (3DNA-based or dendrimer-based nanoparticles) have selectively targeted MG in animal models of PBIs and have shown the benefit of delivering Wnt agonist [91] or miR mimic. One exciting advance has been the discovery that 4 hydroxyl-terminated poly(amidoamine) (G4-OH PAMAM) dendrimers (approx. 4 nm) delivered intravenous cross the BBB and accumulate selectively in activated MG and astrocytes in a mouse inflammatory model [153] and a rabbit model of neonatal HI-injury [154] (but they are not found in glia in the uninjured brain). These PAMAM dendrimers have been used to deliver the antioxidant *N*-acetylcysteine (NAC) in these models to successfully reduce brain injury. Another nanoparticle, a poly (methyl methacrylate) (PMMA) nanoparticles has also been shown to be selectively taken up by MG, and capable of cargo delivery in vivo, but limited to tissue direct delivery (injection into the spinal cord) [155]. Interestingly, the uptake of these PMMA nanoparticles was inhibited in a dose dependent manner by clathrin-mediated endocytosis inhibitor, chlorpromazine (CPZ). Work from this team has also explored further the physical characteristics of PMMA, selective uptake by MG and delivery of minocycline to improve outcome in a spinal cord injury model [156]. Finally, our team has described a MG-specific targeting 3DNA nanoparticle (approx. 200 nM) with no identified toxicity, and no uptake by peripheral macrophage populations in vivo [91]. We have used this 3DNA nanoparticle to deliver an immunomodulatory peptide and demonstrate neuroprotection in a model of systemic-inflammation driven EoP. A highly desirable characteristic of the 3DNA nanoparticles is that they can be delivered intraperitoneal, crossing the BBB to the brain.

Science advances when it can accurately and consistently communicate its ideas. A limitation in the field of MG biology currently is that we lack a clear definition of what markers (enzymes, mRNA species, receptors, cytokines etc) are the core components of the phenotype(s) with functional significance in PBI. This issue is complex, including that there can be expected to be differences in the core members of a phenotype that is injurious in response to different infectious agents [157,158] or even in the white or grey matter located MG [159,160] (for examples) can be different. Standardized nomenclature has benefited many fields, including cell death [161], and recently a consortium has collated an approach for astrocyte biology and there is move underway to derive census in the field of MG biology. Because MG are heterogenous treatments applied systemically will need to target the shared core regulators of function—either good or bad—to have any chance of improving the outcomes for infants with PBI. 

Based on the more recent understanding of the role of metabolism in the maturation and ‘activation’ of MG [162,163] different pathways and potential molecular targets have been proposed as ways to reprogram MG toward beneficial functions [110] (Table 2). However, such molecular reprogramming may be beneficial or detrimental depending on the nature of the injury and timing of intervention, warranting careful consideration when designing experimental paradigms and interpreting data. An example of this is illustrated by the selective liposomal delivery of the antioxidant, reduced glutathione (GSH), to MG in the rat prior to modelling NS at P7 [103]. Although GSH is considered neuroprotective due to its antioxidant activities, pre-treatment increased MG superoxide production with concomitant upregulation of scavenger receptor CD36, in turn, exacerbating neuronal cell death when assessed at 24 h post-NS. Upregulation of CD36 is associated with the anti-inflammatory MG phenotype and mediates phagocytosis of apoptotic bodies following PBI and during normal brain development [164]. However, the respiratory driven process of CD36-mediated phagocytosis leads to generation of ROS, particularly superoxide [165]. And so, while the redox-signaling balance is suggested to modulate polarization of MG phenotypes [164,166], untimely polarization may be ineffective or detrimental to injury outcomes.

## 8. Perspectives and Potential Relevance for Other Brain Disorders

The Kennard Principle describes the concept that the immature brain is more resilient to injury compared to the mature brain and can reorganize and re-structure more effectively to preserve function, and thus is more plastic. However, in the instance of preterm born infants, a great number suffer from permanent neurodisability. This is even though over the past decades’ injury has reduced markedly in severity, now typically being characterised by mild axonopathy and connectivity deficits (for review see [187] and also in TBI [188]). The popularisation of the concept of ‘innate plasticity’ in the immature brain is an impediment to the transfer of information between the fields of perinatal and adult neurodegeneration, as it suggests that the two systems are too different to compare. However, babies do not get better with time (perhaps even the opposite, see text on the tertiary phase) and PBIs share several cellular and molecular features with other brain disorders such as ASD, neurodegenerative disorders or TBI (Table 3). The field of PBIs is greatly benefiting from the knowledge acquired in these other fields. We believe that insights gained in the PBI field might also be of interest due to shared mechanism eventually leading to the identification of targets to boost post-lesioned plasticity in all forms of brain injury situations.

## 9. Conclusions

The specific identity of MG as a unique cell type [189], their roles in PBIs [85,94,103,113,142,190,191,192,193,194,195,196] and novel in vivo MG targeting tools [83,147,148] have emerged over the last decade. Future studies will need to address the precise roles of MG in these disorders by systematically ablating these cells in relevant animal models, despite the complexity to achieve this goal in newborn animals as described above. In addition, work on basic MG development and biology has shown that there is significant regional and temporal heterogeneity in MG [43,153,156,191] that is largely unexplored in models of perinatal brain injury. These effects need to be fully explored not only in rodent models, but in ‘big brain’ models and in human tissues to identify the core processes occurring across regions and time that we need to suppress or support to achieve our aim of improving outcomes for babies who suffer PBI. Finally, once we have our targets, specifically targeting MG with nanoparticles, if they prove to be safe, could represent a major step forward improving the outcome of PBIs.

## Figures and Tables

**Figure 1 biomolecules-11-00099-f001:**
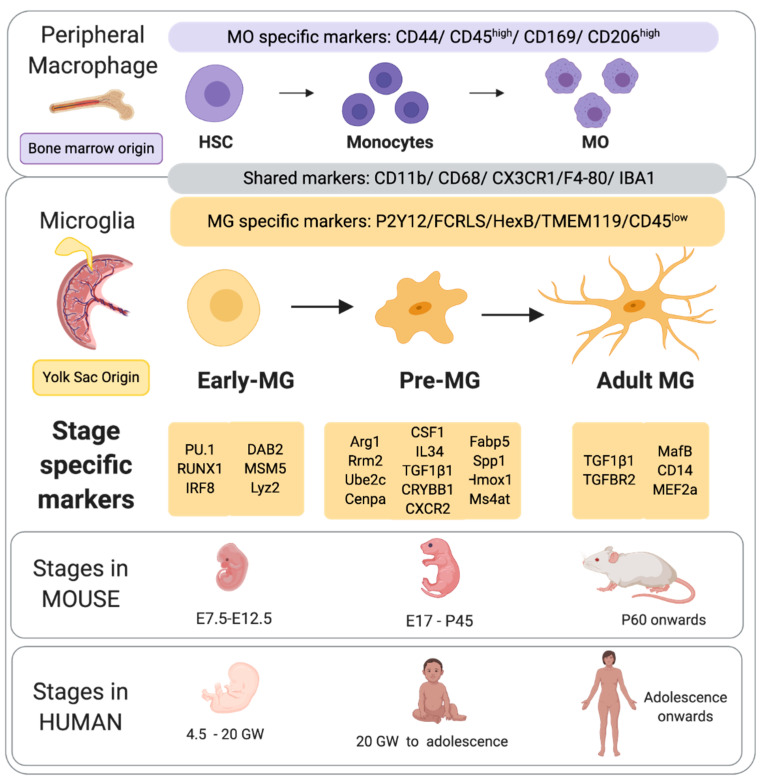
Outline of the origins or macrophage and microglia highlighting common and unique markers. Also, the stage specific markers as drawn and adapted from [35,36,37,38] and stages of maturation of microglia, their approximate timing in mouse and human [39,40]. Created with BioRender.com.

**Figure 2 biomolecules-11-00099-f002:**
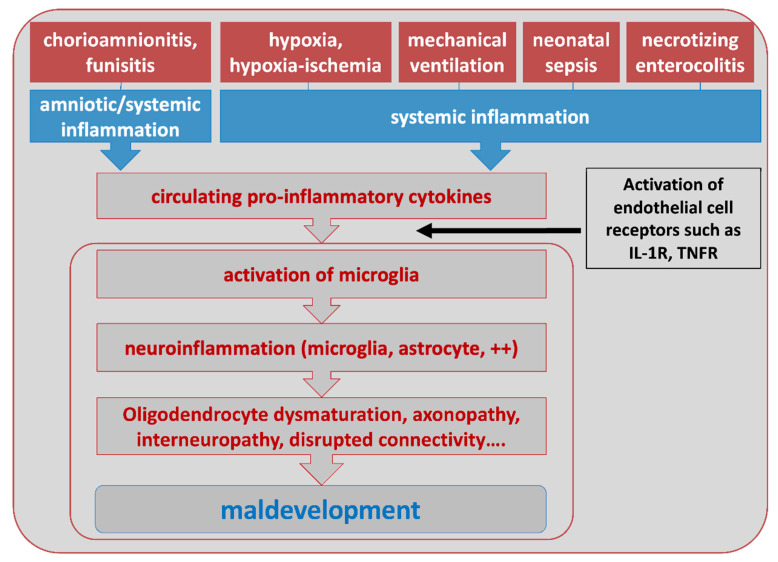
Potential sources of systemic inflammation reported in PBIs and a basic outline of the steps to brain maldevelopment, including from [71,72,73] and others in this review.

**Figure 3 biomolecules-11-00099-f003:**
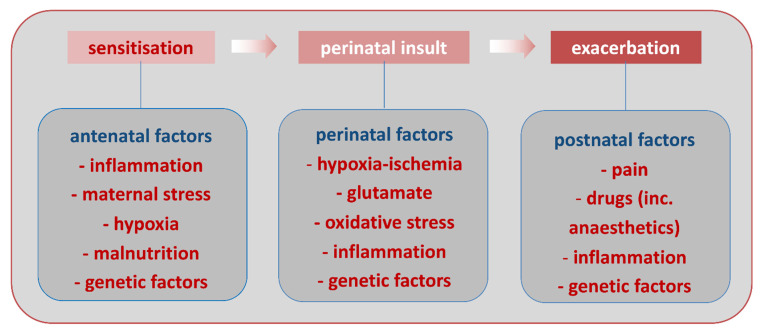
Schematic representation of the multiple-hit hypothesis in PBIs, based on citations from within [21,81] and others in this review.

**Figure 4 biomolecules-11-00099-f004:**
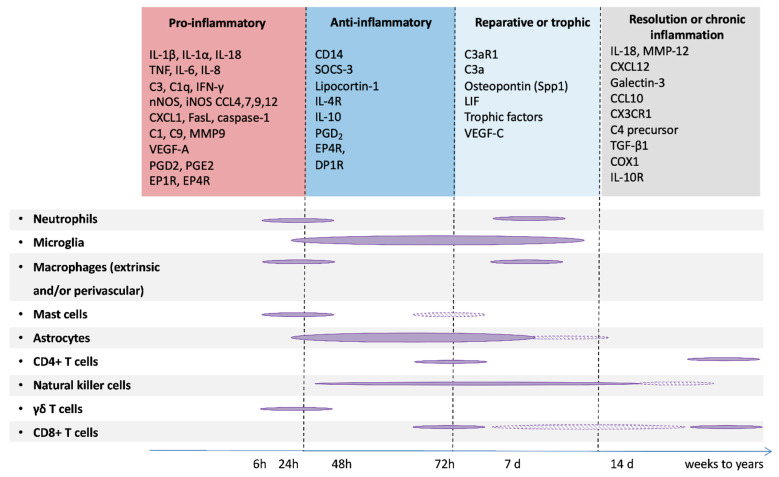
Schematic representation of the immune reaction following a hypoxic-ischemic brain insult in newborn rodents. The hypoxic-ischemic insult triggers a proinflammatory response followed by anti-inflammatory and reparative phases. These events result either in resolution of inflammation or in chronic inflammation. Filled clouds correspond to demonstrated events and shaded clouds correspond to potential events. Adapted from [72].

**Figure 5 biomolecules-11-00099-f005:**
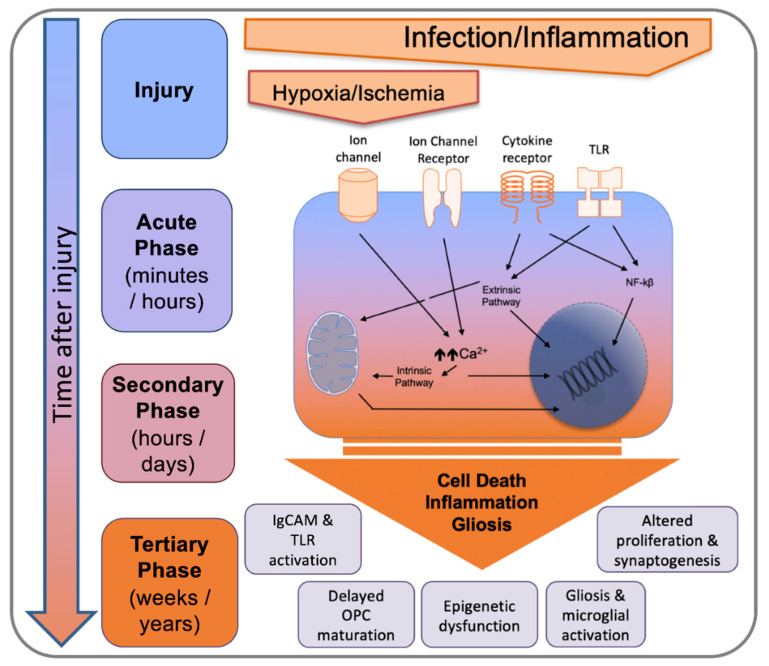
Outline of the acute, secondary, and tertiary damage phases in PBIs. Activation of an array of receptors and ion channels leads to NF-κβ activation, intrinsic and extrinsic apoptotic cascades, and an increase in intracellular calcium, which in combination affect cell death, perpetuate inflammation, and induce gliosis. Adapted from [17].

**Table 1 biomolecules-11-00099-t001:** Brain structures predominantly affected in MRI and/or post-mortem studies of PBIs.

PBIs	Key Features (by Severity)
EoFGR [19]	Reduced cortical grey matter volume Altered cortical gyrification Impaired myelination Reduced connectivity
EoP [22,23]	Diffuse white matter injury (including cerebellum) Punctate white matter cystic lesions Reduced grey matter microstructure Reduced connectivity (in particular thalamocortical connectivity) Intra-ventricular hemorrhages
NE [22]	Destructive lesions of cortical areas Destructive lesions of basal ganglia Impaired myelination of the posterior limb of the internal capsule (PLIC)
NS [20]	Destructive lesions of arterial or venous territories

**Table 2 biomolecules-11-00099-t002:** Potential therapeutic targets to modulate MG effects in PBIs. Adapted from [104].

Molecular Targets	Candidate Interventions
PPARγ receptor	Pioglitazone, Rosiglitazone, Metformin [167,168]
AMP-related kinase (AMPK)	Metformin [169]
Aldose reductase	Sorbinil, Zopolrestat, Fidarestat, Tolrestat [170,171]
Sirtuin 1	Resveratrol [172]
Wnt pathway	Wnt pathway agonists (via 3DNA nanoparticle delivery) [91]
MicroRNAs (miRs)	miR mimics and antago-miRs (via 3DNA nanoparticle delivery) [110]
Omega3-omega6 fatty acid balance	Enriched omega3 fatty acid nutrition [173,174]
Steroid receptors	Estetrol (E4) [175]
Cyclo-oxygenase-2 (COX)-2	Nimesulide, Indomethacin [133,176]
Leukotriens	Montelukast
Microbiota and gut	Prebiotics, probiotics, faecal transplant [177,178]
Multiple targets	Mesenchymal stem cells [179,180,181], melatonin [182]
Redox balance, *Nrf2*	GSH: GSSG, N-acetyl-cysteine, sulforaphane [183], p38 [184]
Transforming Growth Factor Beta	ALK5 [185,186]

**Table 3 biomolecules-11-00099-t003:** Potential overlap between PBIs and other brain disorders.

Brain Disorders	Potential Similarities with PBIs
ASDs	Pro-inflammatory MG activation Myelin defects Impaired connectivity Impaired synaptogenesis Impaired microbiota Sex bias
MS	Pro-inflammatory MG activation Reactive astrocytes OPC maturation arrest Myelin defects
PD and AD	Pro-inflammatory MG activation Reactive astrocytes Protracted neuronal cell death Sensitization, systemic inflammation
TBI	Pro-inflammatory MG activation Impaired astrocyte functions Myelin defect Tertiary phase

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
