# Peer review of "Microglia-Mediated Neurodegeneration in Perinatal Brain Injuries"

_biomolecules, 2021, doi:10.3390/biom11010099_

Round 1
Reviewer 1 Report
This review focuses on the role of microglia in the pathogenesis of neurodevelopmental disorders. The authors reviewed all pathologies associated with perinatal brain injuries, such as encephalopathy associated with fetal growth restriction, encephalopathy of prematurity, neonatal encephalopathy of term neonates, and neonatal stroke, in almost all aspects related to the role of microgolia in pathogenesis and brain development. It is crucial that the authors managed to combine both clinical data and the results of studies on experimental models. It should be noted that the review of pathogenetic brain damage during embryonic development or immediately after birth is a complicated task due to the presence of different stages of brain development and many etiological factors preceding brain damage. In this regard, in this reviewer's opinion, for the sake of completeness, it is desirable to describe the stages of microglia development starting from early gestational age (23-25 weeks) and ending with brain development in the early childhood period. These data will better understand the role of microglia in brain damage, especially in cases of encephalopathy associated with fetal growth restriction and encephalopathy of prematurity. It is also necessary to describe in detail the titles to Figure 3 (clarification is required in the designation of shaded and filled clouds) and Figure 4. Table 2 needs proper literature references confirming the therapeutic activity of presented drugs to microglia. In general, the review article is original and useful for developing progress in this area of knowledge.
Author Response
Dear Editorial office and reviewers,
Thank you for your patience while we made these changes over the holidays and we wish you all a safe and happy new year.
We appreciate the opportunity to improve the work and have addressed the comments in detail in the manuscript (visible with track changes) and we will summarise below. We have also gone through the manuscript to correct grammatical errors.
Reviewer 1.
Microglia development. As requested we have included a section on the basic development of microglia (pg 3) and included a figure to outlines these stages (Figure 1)
Figure titles. We have elaborated on the description of the figures (pg 11).
References. We have included relevant references for the validated efficacy of each of the listed therapeutics in Table 2 (pg 9).
Reviewer 2 Report
Fleiss et al. reviewed the role of microglia in various perinatal brain injuries, using both human and animal data. Excellent tables and diagrams help the reader navigate the presented topic. The manuscript is well written, and the storyline is easy to follow because of the clear organization of chapters.
Chapters 1-3 cover the basis of perinatal brain injury, systemic inflammation, and the role of microglia, providing a substantial background for the rest of the manuscript which tackles more specific questions regarding microglia and poses unexpected questions for the audience. The abstract stands alone and seems to convey a different message than the main text. It gets the reader hooked by a provocative statement, but it doesn’t represent all the key topics covered in the manuscript, such as prospective therapeutic interventions targeting microglia. Surprisingly, a bold conclusion of the abstract plays out in a toned-down manner in the manuscript. Despite this discrepancy, the manuscript is insightful and relevant to the field.
A minor adjustment of the abstract or Ch. 7 would increase the integrity of the manuscript.
Author Response
Dear Editorial office and reviewers,
Thank you for your patience while we made these changes over the holidays and we wish you all a safe and happy new year.
We appreciate the opportunity to improve the work and have addressed the comments in detail in the manuscript (visible with track changes) and we will summarise below. We have also gone through the manuscript to correct grammatical errors.
Reviewer 2.
Abstract alignment. We agreed with the reviewer and have both ‘toned down’ the abstract (pg 1) and ‘toned up’ the text to include a little more on the roles of MG and problems with targeting these cells (Sections 4-7).
Round 2
Reviewer 1 Report
The authors have done a good job revising the manuscript. I have no further questions.